# OpenReview forum: "LLM Evaluators Recognize and Favor Their Own Generations"
_NeurIPS.cc/2024/Conference — NeurIPS 2024 oral_

### Official Review · Reviewer_48mP · 2024-07-07

**Soundness:** 3
**Presentation:** 4
**Contribution:** 4
**Rating:** 9
**Confidence:** 3

**Summary:**

This paper investigates whether large language models can identify their own generations in two settings: when they have to distinguish between their output and another output from another large language model or person, and when they are only given an output and must score according to a Likert-scale 1-5. They then investigate the correlation between self-recognition and self-preference. They find their explanation resists some confounders.

**Strengths:**

- They authors find novel results that will significantly influence the training and evaluation of some models
- Simple and clear presentation of methodology, results, and limitations
- Compelling analysis into potential confounders

**Weaknesses:**

No major weaknesses - the paper was a pleasure to read!

Small points:
- L103 should have “GPT-3.5” instead of “GPT-3”
- L413 starts with “??e collect”

**Questions:**

- To compute the final self-preference rating, why did you average the five possible scores weighted by the output probability rather than greedily selecting the score with the highest probability?
- On line 106, you noted “... goes against our intuition that self-recognition scores should increase as the dissimilarity between evaluator and evaluatee increases”. It’s unclear to me whether you still share this intuition after concluding this research?

**Limitations:**

The authors address this.

---

> ### Author Response · Authors · 2024-08-07
>
> Thank you for the feedback on presentation! We will incorporate them in the revision. Here we'd like to respond to the two questions:
>
> > why report weighted average rather than greedily selected quality score
>
> We do so mainly due to the insensitivity of LLMs in the individual measurement setting. We frequently see LLMs assigning the same score to most or all summaries. If we report the greedily sampled score, we will essentially see zero recognition/preference across the board. To get a better idea of the fine-grained differences that the LLMs might pick up, we choose to report the weighted average.
>
> > ...intuition that self-recognition scores should increase as the dissimilarity between evaluator and evaluatee increases
>
> Great question. We are still undecided on this intuition. This specific result can be an isolated phenomenon due to GPT-4’s bias to not recognize anything as written by itself. In the follow-up work, we will expand experiments to a wider range of frontier models and include base models that are less affected by RLHF fine-tuning.
>
> Thanks again for your time!

---

> > ### Comment · Reviewer_48mP · 2024-08-12
> >
> > I accept the first argument. I am excited to see this follow-up work.

---

### Official Review · Reviewer_gQGf · 2024-07-10

**Soundness:** 2
**Presentation:** 3
**Contribution:** 2
**Rating:** 5
**Confidence:** 3

**Summary:**

Self-evaluation is widely adopted but can lead to self-preference in that the LLM evaluator scores its own outputs higher than others while the qualities are actually equal. This paper finds that LLMs prefer their own generation because they recognize themselves. This paper conducts experiments on two summarization tasks and 3 LLMs. First, the authors show that LLMs exhibit self-preference in self-evaluation and they have a good ability in self-recognition. Then, the authors fine-tune the LLMs to make the ability of self-recognition almost perfect. They find that the self-preference strength is linearly correlated with self-recognition. The authors conduct various experiments to avoid confounding from the quality differences, ordering, fine-tuning improving quality, and other confounders from fine-tuning. Finally, the authors mention two safety concerns related to their findings.

**Strengths:**

1. This paper is well-written and easy to follow. All the details are shown in either the main paper or the appendix.

2. The authors sufficiently mention previous works and clearly claim their own contribution against them, without over-claiming their contribution.

3. The authors intentionally design more supplementary experiments and analyses to rule out the confounder.

**Weaknesses:**

1. Correct me if I am wrong. I think the hypothesis of this paper is problematic.  It is not "an LLM prefers a sentence due to it is generated by itself", but "an LLM generates a sentence due to it prefer it". As for your experiment on increasing self-recognition leads to the rise of self-preference, please refer to weakness 2.

2. Correct me if I am wrong. I think there is another confounder in fine-tuning. Is there any chance that the fine-tuning lets the model learn a shortcut that selects the labeled one? For example, using self-recognition data to fine-tune an LLM may make the LLM overfiting to select the self-recognized one even though it is asked to select the high-quality one. This potential issue can be eliminated by checking if the LLM is more likely to select the shorter one as the high-quality one after fine-tuning on the task of selecting the shorter one.

3. The scope is limited. This paper only conducts experiments on 2 summarization datasets and 3 LLMs, leading to the concern of the generalization of their findings.

**Questions:**

1. In line 31, you mention "Is self-preference truly self-preference, in the sense that the LLM prefers a text because it was generated by itself?". Do you think it is the opposite: "LLM generates a text because it prefers it"? If so, I think the main hypothesis of this paper is problematic. It is not "an LLM prefers a sentence due to it is generated by itself", but "an LLM generates a sentence due to it prefer it".

As for your experiment on increasing self-recognition leads to the rise of self-preference, as I mentioned in weakness 1, it may be due to the shortcut learned during fine-tuning. I strongly recommend you to conduct another experiment, such as checking if the LLM is more likely to select the shorter one as the high-quality one after fine-tuning on the task of selecting the shorter one.

2. Besides, there is an easier way to verify if LLM prefers their own generation because they recognize themselves. We can directly tell them which one is generated by themself and which one is not. And then ask them to evaluate which one is better. If the self-preference ratio increases, we can conclude that LLM prefer their own generation because they recognize themselves.

3. Experiments on 3 LLMs on 2 summarization benchmarks are too small to convince me of the generalization of the findings.

4. Typos:
 Line 413: ??

I will adjust my rating if you can address my concerns.

**Limitations:**

There is no potential negative societal impact. The authors have discussed the limitations.

---

> ### Author Response · Authors · 2024-08-07
>
> Thank you for the careful review. The two experiments you suggested are in fact already in the paper (please let us know if there are better ways to highlight them). Below we provide a response which should hopefully clear up the confusion.
>
> > Correct me if I am wrong. I think the hypothesis of this paper is problematic. It is not "an LLM prefers a sentence due to it is generated by itself", but "an LLM generates a sentence due to it prefer it".
>
> This is incorrect. Since all the summaries are generated by LLMs without reference summaries, the hypothesis "an LLM generates a sentence due to it prefer it" would not even comply with the generation process.
>
> > Correct me if I am wrong. I think there is another confounder in fine-tuning. Is there any chance that the fine-tuning lets the model learn a shortcut that selects the labeled one?
>
> This is incorrect. First, the fine-tuning examples are distinct from the evaluation examples, and the ordering of the two options are randomized & balanced for both, so the increase in self-preference cannot be trivially explained. If by “overfitting” and “shortcut learned during fine-tuning” you are referring to the phenomenon that in self-preference evaluation, the model recognizes patterns it learned about its own generation in fine-tuning, and that causes more bias—this would exactly in line with our causal hypothesis that self-preference is caused by recognition of itself.
>
> We are not trying to explain *how* the model is recognizing itself; our hypothesis is focused on its effect on self-preference. We only control for confounders between these two properties, not shortcuts that the model can use for self-recognition. Note that the experiment suggested by the reviewer is already in the paper—fine-tuning to select a shorter/longer summary from the pair (Section 3.4)—along with many other control tasks. This did not significantly affect either self-recognition or self-preference strength.
>
> > Limited scope
>
> We agree that experimenting on more tasks is worthwhile and will update the paper once more experiments are completed. The following are our primary considerations for focusing on the two summarization tasks as the first step.
>
> 1. Summarization is a representative testbed for self-evaluation. As one of the first tasks used to demonstrate RLHF, summarization remains one of the most mature testbeds of LLM self-evaluation for long-form generation. With comprehensive scoring guidelines and human-written references, summarization fits our use case perfectly.
> 2. Cost. Our experiment—like most involving frontier LLMs—is costly, particularly due to the number of control tasks included in the fine-tuning experiment. Not counting the Cloud compute costs for Llama experiments, we used roughly 300 million tokens via inference API for evaluation, and 150 million tokens for fine-tuning.
> 3. Diversity. Although both XSUM and CNN/DailyMail are summarization tasks which follows the same evaluation protocol, the difference between them—extractive vs. abstractive summarization—improves the diversity of the evaluation and provides evidence for the generality of self-recognition/self-preference, as analyzed in Sec 3.2 and Figure 7.
>
> > We can directly tell them which one is generated by themself and which one is not. And then ask them to evaluate which one is better.
>
> We have conducted this exact experiment in Section 3.5, where we label the source of each piece of text and re-evaluate self-preference. This indeed leads to increase in self-preference as you hypothesized. We further experimented with using intentionally incorrect labels. This is to check if the model is capable of recognizing the real generation from textual features even when we “lie” to them in the prompt, and indeed, especially for GPT-4 on the CNN/DailyMail dataset, the model still strongly prefers its own generation (the real one) when we lie to GPT-4 that the other summary is generated by it.
>
> Thanks again for the careful review. Hopefully this addresses your concerns. Let us know if there is any other clarification we can provide.

---

> > ### Comment · Reviewer_gQGf · 2024-08-13
> > **Thanks for the response.**
> >
> > Thanks for the response. I will raise my assesment.

---

### Official Review · Reviewer_g5q5 · 2024-07-10

**Soundness:** 3
**Presentation:** 3
**Contribution:** 2
**Rating:** 6
**Confidence:** 4

**Summary:**

The paper investigates the novel topic of self-preference and self-recognition in large language models (LLMs). The experiments are well-conceived, and the use of pairwise comparison alongside individual evaluation provides a solid framework for understanding these phenomena. Despite these strengths, the work suffers from significant limitations, including a lack of sufficient experimental diversity and statistical rigor, which undermine the overall impact and reliability of the findings.

**Strengths:**

The idea is novel and addresses an important issue in the field of AI.
The use of both pairwise comparison and single input individual evaluation is a thoughtful approach that offers valuable insights into the behavior of LLMs.
The approach of fine-tuning LLMs to investigate self-recognition and self-preference is innovative and provides new insights into model behavior.

**Weaknesses:**

The experiments are too controlled with minimal variety, lacking sufficient breadth to thoroughly explore the randomness of the models. Additional experiments, particularly with pairs that do not include self-generated text, would strengthen the findings.

Figures, especially Figure 2, are not clearly explained. The paper would benefit from more detailed descriptions to help readers understand what each figure represents.

The paper lacks comprehensive statistical results to support its claims. More robust statistical analysis is necessary to validate the findings.

The study does not consider pairs without self-created summaries, which could provide crucial insights into whether LLM preferences are genuinely self-preferential or random.

**Questions:**

What do you think will happen if any text generated by the LLM is then paraphrased by using some paraphrasing tool and then calculated the self-recognition score? Do you think LLMs will still be able to identify that particular text as text generated by them?

Corrections:

Line 163: the the - >  the

Line 215: need - > needed

Line 252: use - > used

Line 288: self-recognitiono -> self-recognition

Also one of the paper is cited twice, the 3rd and 4th papers are the same.

---

> ### Author Response · Authors · 2024-08-07
>
> Thank you for the thoughtful feedback! We'd like to address the weaknesses and questions you brought up.
>
> > Limited scope
>
> Within the context of summarization and the constraints with compute budget, we maximized the coverage in the following aspects of the experiments, to the best of our ability:
> 1. Task diversity: by experimenting on both extractive and abstractive summarization
> 2. Evaluation format: pairwise and individual
> 3. Fine-tuning setup: in- and out-of-domain, number of training examples
> 4. Control tasks: to rule out as many confounders as we can
> 5. Ordering of options in pairwise evaluations
>
> > Statistical significance
>
> We perform Chi-Squared tests and confirm the statistical significance of all the following claims (p-value << 0.001):
> 1. LLMs demonstrate preference for their own generations disproportionately compared to humans
> 2. LLMs demonstrate significantly higher self-preference after fine-tuning for self-recognition
> 3. There is a significantly higher increase in self-recognition and self-preference from fine-tuning for self-recognition compared to fine-tuning on the control tasks.
>
> We will update the draft with these more comprehensive details.
>
> > Non-self-created baseline
>
> We agree that a baseline of pairwise evaluation on non-self-created texts exclusively is a good addition, and will incorporate that in the camera-ready version. We do note that the existing pairwise results already address the ordering bias by evaluating each pair twice, with both orderings of the options. In addition, the individual measurements that demonstrate negligible self-preference suggest that the LM will likely (correctly) assign close to 50-50 when neither example in a pair is written by itself (assuming equal quality).
>
> We appreciate your feedback on presentation of figures and will incorporate it in revision. Thank you!

---

> > ### Comment · Reviewer_g5q5 · 2024-08-09
> >
> > Thanks you for the considerations, with the following additions made to the camera-ready paper, I am increasing my score to 6.

---

### Official Review · Reviewer_jTdx · 2024-07-11

**Soundness:** 3
**Presentation:** 4
**Contribution:** 3
**Rating:** 7
**Confidence:** 4

**Summary:**

The authors examine self-preference in language models through the lens of self recognition. They pose the following question: if models indeed prefer themselves, is it also because they recognize themselves? The authors explore a range of models and find correlates between self recognition and self-preference. Furthermore, the authors explore potential causal links between self-recognition and preference by finetuning models on confounding tasks (a true causal analysis is prohibitive given that a mechanistic understanding of LLMs is unavailable).

**Strengths:**

Disentangling preference and recognition is an insightful idea! This enables the authors to explore a potential causal relationship.

Cross-checking with a human evaluation is great; so is taking into account ordering effects. It’s clear that the authors put thought into their prompting and evaluation.

I think the control tasks are quite diverse, and the paper does a good job analyzing potential confounds.

**Weaknesses:**

Some weaknesses are formulated as questions in the questions section.

I do have one (big-ish) concern. Summarization is a limited task. The authors do mention this in the limitations of the paper—but a potential confound might be memorization of the summarization datasets (e.g. models that have seen the dataset more during training are also more likely to prefer the same dataset). I really do think it is worthwhile running preliminary expts. on other domains.

**Questions:**

While the paper is well written and comprehensive, I have a few questions I'd be curious about:

1. Scaling effects: Are larger models better at self-recognition? I would’ve liked to see trends across scale. I was looking at Figure 1 and trying to understand if there was a trend (e.g. GPT 3.5 is purportedly smaller than 4), but it would’ve been nice to see Llama 70B results. If this isn’t possible due to computational constraints, I totally understand! But some hypotheses would still be nice.

2. Are differences in preferences between LLMs and human annotators statistically significant? Re: this line, the authors claim significance- is there a test that backs this up?

```But the disparity between LLMs as rated by humans is significantly lower than the level of self-preference exhibited by the LLMs, in particular GPT-4. This suggests that out of the box, the LLMs’ self-preference is disproportionate to the actual quality differences.```

3. How much of this goes away with a prompting mitigation? (e.g. include something like “Don’t be biased to your own outputs”) in the prompt? I think that would be a really interesting finding—regardless of what you find.

A small suggestion: in Figure 1, I would draw a vertical / horizontal line at x = 0.5 and y = 0.5, just to quickly see which models fall in which quadrants.

Line 175: it’s -> its

**Limitations:**

Yes

---

> ### Author Response · Authors · 2024-08-07
>
> Thank you for the thoughtful feedback! We'd like to address some concerns and questions brought up in the review.
>
> > Limitations of running the experiments on two summarization tasks
>
> We agree that experimenting on more tasks is worthwhile and will update the paper once more experiments are completed. The following are our primary considerations for focusing on the two summarization tasks as the first step.
>
> 1. Summarization is a representative testbed for self-evaluation. As one of the first tasks used to demonstrate RLHF, summarization remains one of the most mature testbeds of LM self-evaluation for long-form generation. With comprehensive scoring guidelines and human-written references, summarization fits our use case perfectly.
> 2. Cost. Our experiment—like most involving frontier LLMs—is costly, particularly due to the number of control tasks included in the fine-tuning experiment. Not counting the Cloud compute costs for Llama experiments, we used roughly 300 million tokens via inference API for evaluation, and 150 million tokens for fine-tuning.
> 3. Diversity. Although both XSUM and CNN/DailyMail are summarization tasks which follow the same evaluation protocol, the difference between them—extractive vs. abstractive summarization—improves the diversity of the evaluation and provides evidence for the generality of self-recognition/preference, as analyzed in Sec 3.2 and Figure 7.
>
> > Scaling effects
>
> Based on our experiments, larger and more capable models do appear better at self-recognition. One hypothesis is that larger models’ output probabilities are more focused around their own generations. State-of-the-art LLM detection methods compute the LLM’s perplexity on generated text vs. perturbed versions and use that gap for detection. Given evidence that suggests that detection is at least not more difficult for larger and more capable LLMs, we hypothesize that larger LLMs’ perplexity is more sensitive to whether the text is generated by the LLM. If self-recognition relies on similar mechanisms as detection methods (this is one hypothesis we are investigating), then this might explain why larger LLMs are better at self-recognition.
>
> > Significance of difference in preference between humans and LLMs
>
> Using the pairwise format, we run a Chi-squared test of statistical significance for the difference between self-preference (Figure 4) and human preference (Sec 2.5), and find that the difference is significant (p-value << 0.001) for GPT-4 (compared to GPT-3.5 and Llama 2) even prior to fine-tuning, and significant (p-value ~ 0.007) for GPT-3.5 after fine-tuning on 10 examples for self-recognition ability. We will update the draft with these more comprehensive details.
>
> > Prompt sensitivity
>
> In our initial experiments, self-recognition/self-preference seems insensitive to instructions in the prompt. We refrain from prompts like “Don’t be biased to your own outputs” in self-preference evaluation because we don’t have a good way to decouple the effect of priming the model to think that one of the inputs is from itself. For this submission, we wanted to stick to the main message so as to not confuse readers. For follow-up work we will perform more thorough prompt engineering, including giving GPT-4 a better prior of the likelihood of its own outputs showing up.
>
> Thank you again for your time. Let us know if there is any other clarification we can provide or if you have other suggestions!

---

> > ### Comment · Reviewer_jTdx · 2024-08-12
> > **Thanks!**
> >
> > Thanks for the rebuttal! I'm keeping my (positive) score, since the current paper + rebuttal only has results on summarization. Still, great work!!

---

### Decision · Program_Chairs · 2024-09-25

**Decision:**

Accept (oral)

**Comment:**

I am very certain that this paper should be accepted and as it is the highest rated paper in my cohort I am recommending it for the highest level of acceptance which would be an oral presentation.  The main reason for this is that I believe that the tendency to use an LLM as a judge when evaluating an outcome is quite common and the authors here surface an important bias of LLMs to favor their own generations which strongly implies that GPT-4 would not be an impartial judge between outcomes form a GPT4 pipeline vs another LLM pipeline.  Given the importance of this bias for many I am recommending oral so that this gets the most exposure.  The key weakness, surfaced by the reviewers is that only summaries were tested.  In light or this weakness it may be that the presentation could be downgraded, but I feel strongly that it should be accepted in some form.